# Thermal and Quantum Fluctuation Effects on Non-Spherical Nuclei: The Case of Spin-1 System

**Mohammed Mahmud** [1,2,*] , **Mulugeta Bekele** [2,*] and **Yigermal Bassie** [3]

[1] Department of Physics, Wallaga University, Nekemt P.O. Box 395, Ethiopia
[2] Department of Physics, Addis Ababa University, Addis Ababa 1000, Ethiopia
[3] Department of Physics, Wolkite University, Wolkite P.O. Box 07, Ethiopia
[*] Correspondence: mohammedmahmud1@gmail.com (M.M.); mulugetabekele1@gmail.com (M.B.)

**Abstract:** In this work, we investigate the collective role of thermal and quantum fluctuations on non-equilibrium thermodynamics of a quantum system, specifically, the quantum-thermodynamic description of spin-1 nuclei based on the concepts of quantum and statistical mechanics. We explore the dynamical response of the system when driven out of equilibrium by a work parameter and compute analytically the full distribution of the work generated by the process. Considering work performed on the system as a random variable, we collect data for a large number of repeated cyclic processes of finite time. These data of finite time non-equilibrium processes will permit us to derive equilibrium values in quantities such as the free energy difference between the final and initial states of the system. Various properties of the system's work distribution are explored.

**Keywords:** nuclear magnetic resonance; non-equilibrium process; spin-1 magnetic system; quantum thermodynamics

## 1. Introduction

In the real physical world, both thermal and quantum fluctuations play a crucial role in bringing change to the state of a system itself or to the system's state. This implies that the collective contribution of thermal and quantum fluctuations brings out changes in the thermodynamic properties of matter. Fluctuations are also important on nanometer scale systems and lead to the large variability of mechanical and functional properties. They may also create noises affecting the performance of the devices. Thus, fluctuations are unavoidable either in the thermodynamic description or in the atomic description of the system. The field of study that incorporates both thermal and quantum fluctuations in its description of the system is known as quantum thermodynamics.

Quantum thermodynamics emerges as a theory aimed to interrelate the microscopic theory to macroscopic measurements. This means that it plays a continuous dialogue between the two independent theories: thermodynamics and quantum mechanics. So, in the study of quantum thermodynamics, it is possible to address the thermodynamic laws from inherent quantum mechanical theories. Unlike that of quantum statistical mechanics [1], it emphasizes the dynamical processes out of equilibrium. Hence, in quantum thermodynamics, there is a quest for the study of a single individual quantum system. The recently developed, highly controlled quantum experiments ([2,3]), the availability of powerful numerical methods [4], and the development of novel theoretical tools ([5,6]) further help scientific scholars rely on the field of quantum thermodynamics. In general, quantum thermodynamics is used to extend the theory of standard thermodynamics [7] and non-equilibrium statistical mechanics [8] to an ensemble of sizes well below the thermodynamic limit, in non-equilibrium situations, and with the full inclusion of quantum effects.

In thermodynamics, as a macroscopic theory [7,9], the limited number of macroscopic variables, such as volume, temperature, and pressure are used to completely characterize the state of the thermodynamic system near equilibrium, and further used to model the

exchange of energy, work and heat based on few laws. Equilibrium thermodynamics is the systematic study of the transfer of matter and energy within the system as they pass from one equilibrium state to another. On the other hand, non-equilibrium thermodynamics deals with physical systems that are not in thermodynamic equilibrium in terms of variables that represent an extrapolation of the variables used to specify the system in thermodynamic equilibrium. Non-equilibrium thermodynamics is mainly concerned with transport processes and with the rate of chemical reactions [10].

The microscopic constituents of the thermodynamic system are always in a state of inherent chaos and randomness so the variables underlying the microscopic world are constantly fluctuating. Such fluctuations in non-equilibrium dynamics contribute to quantum fluctuations. The out-of-equilibrium dynamics of quantum systems has so far produced important statements on the thermodynamics of small systems undergoing quantum mechanical evolutions [11,12]. Particularly, Crooks and Jarzynski's relations [13,14] have provided the relations that connect equilibrium properties of thermodynamical relevance with explicit non-equilibrium features. Although in the real world, it is impossible to isolate a particular quantum system from its surroundings, recent advancements in experimental techniques allow one to measure and control systems at the level of single molecules and atoms [15,16]. The random fluctuations present in small systems affect thermodynamic quantities, such as work and heat, so they must be included in the description of the whole system's dynamics. Recently, a number of authors have proposed definitions of work, and derived fluctuation theorems for quantum systems [5,13,17–19]. However, in non-equilibrium processes, the work conducted must be treated as a random variable [14,20]. Therefore, a statistical approach is needed to find the average value of the work distribution.

The Gibbs formulation in equilibrium statistical mechanics [21] provides a powerful means to explain the macroscopic properties of thermodynamic systems from the fluctuating microscopic variables. Since the quantum fluctuations (ever existing at the atomic level) cannot be easily measured macroscopically, statistical mechanics focuses on the ensemble of a large number of systems where those fluctuations become negligible. So, the measurements we experience at the macroscopic level take place by averaging the fluctuations taking place at the microscopic level. The averaging is performed by implementing the law of large numbers [22]. Indeed, talking about the average behavior of a given ensemble of systems in statistical mechanics is probabilistic.

Studies on the non-equilibrium thermodynamics of nuclear magnetic resonance (NMR) of spin-half sample nuclei have been performed in 2016 [19]. We extend their work to the NMR of spin-one nuclei where the sample atomic nuclei are first subjected to a strong static magnetic field, $\mathbf{B}_0$, in the z-direction and are perturbed by a weak radio frequency field, $\mathbf{B}_1$, applied along a direction perpendicular to $\mathbf{B}_0$. As the nuclei interact with $\mathbf{B}_0$, their energy splits into three levels with respect to the orientation of $\mathbf{B}_0$. Then, $\mathbf{B}_1$, causes a transition in the state of the system from its previous state. So, we intend to solve the work distribution as transitions are promoted by the perturbation. We further want to monitor the average work conducted both as a function of frequency and time. In general, Spin-1 nuclei such as deuterium atomic isotopes when exposed to an external magnetic field will orient in three possible discrete states. To the best of our knowledge, we have not seen any previous work dealing with their quantum thermodynamic properties. Their rich properties motivated us to address them.

The rest of this work is organized as follows. In section 2, the model system is presented along with the protocol to administer the cyclic process in order to collect data. In Section 3, the time evolution of the system under static and rotating magnetic fields is worked out. Section 4 derives the expression for the mean polarization along the $x$, $y$, and $z$-axis. Section 5 deals with the different aspects of work distribution. In Section 6, work distribution properties are explained starting from their characteristic functions. Lastly, the summary and conclusion are given in Section 7.

## 2. The Model

We consider a sample of spin-1 nuclei placed under a strong static magnetic field, $\mathbf{B}_0$, oriented in the z-direction. Since in such a nucleus the charge distribution is not spherically symmetrical, the energy of the nucleus depends on its spin orientation with respect to the non-uniform internal electric field existing at its location [23]. Accordingly, each nucleus can be in any one of the three quantum state orientations: parallel, anti-parallel or perpendicular to the external magnetic field, $\mathbf{B}_0$. The spin Hamiltonian for the interaction of every single spin with this static field, $\mathbf{B}_0$, is given by

$$H_0 = -\boldsymbol{\mu} \cdot \mathbf{B}_0 \tag{1}$$

where $\boldsymbol{\mu}$ is the magnetic polarization of the nuclei. The nuclear spins are more strongly coupled to the external environment than to their molecular environment. So, by neglecting the effect of quadrupolar interactions of the nuclei, which is due to the rotation of the electric charge of the nuclei, here we only consider nuclear spin interaction with the external magnetic field. We attached this spin-1 system to a heat bath of temperature $T$ while we made an assumption that the system is extremely weakly coupled to the bath such that the energy relaxation and the decoherence rate are proportionally very small for the undergoing process to be unitarily evolving. Initially, we let the system equilibrate with the heat bath. A finite-time cyclic process was carried out according to the following procedure with a corresponding collection of data during each cycle.

Model of operating the cyclic process and collecting data—the protocol. Once the system has equilibrated, a weak alternating r.f. magnetic field is switched on and its corresponding energy value, $E_i$, is measured at the start time ($t = 0$). The r.f. field is imposed to act on the system until time $\tau$ and switched off while its corresponding energy value, $E_f$, is measured at the end. The system is allowed to equilibrate once again and the same procedure of switching on and off the r.f. magnetic field is carried out with its corresponding measurements of $E_i$ and $E_f$ taken. This cyclic process is performed a large number of times up until there are enough data to carry out the statistics.

## 3. Time Evolution of the Model under Static and Rotating Magnetic Fields

In this section, we place our model system in a strong static magnetic field $B_o$ in the z-direction and study its time evolution using the standard Schrodinger equation (Section 3.1). Our system is further subjected to a weak alternating r.f. field rotating in the xy-plane with frequency $\omega_z$ and the time evolution of the combined action is studied accordingly (Section 3.2).

### 3.1. Spin-1 in a Static Magnetic Field

The Schrodinger equation for a nuclear spin in a strong static magnetic field oriented along the z-direction is:

$$i\frac{\partial |\psi\rangle}{\partial t} = \hat{H}_0 |\psi\rangle = -\boldsymbol{\mu}_z \cdot \mathbf{B}_0 |\psi\rangle = -\omega_0 \mathbf{I}_z |\psi\rangle. \tag{2}$$

where we have used $\omega_0 = -\gamma B_0$ as Larmor frequency, $\gamma$ the gyromagnetic ratio and $\hbar = 1$. Since the Hamiltonian, $H_0$, is proportional to an operator ($\mathbf{I}_z$), $H_0$ and $\mathbf{I}_z$ commute and as a result share common eigenstates. This statement will be clear when we write the Hamiltonian as a matrix in the Zeeman eigenbasis of $\mathbf{I}_z$. Because the static magnetic field considered is uniform, the orientation of the spin changes periodically. This means, if it is initially oriented along the z-direction, it periodically returns to that direction. Since Equation (2) is the time-independent Schrodinger equation, its solution will be

$$|\psi(t)\rangle = e^{(i\omega_0 \hat{I}_z t)} |\psi(0)\rangle. \tag{3}$$

This implies that if the state of the spin system is known at one point in time, then it is possible to predict it at later times by applying the time-dependent Schrodinger equation to each individual spin. Accordingly, for the spin-1 nuclei coupled to the heat bath at inverse temperature, $\beta$, the state of the system can be expressed by the Gibbs density matrix as

$$\rho_{th} = \frac{e^{-\beta \hat{H}_0}}{Z_0}, \tag{4}$$

where $Z_0$ is the initial partition function which ensures the normalization condition of the state density matrix.

Let $E_n^0$ and $|n\rangle$ be the eigenvalues and eigenvectors of the Hamiltonian $H_o = H(\lambda_0)$. Then, the probability $P_n$ of having the system in the state $|n\rangle$ with energy eigenvalue $E_n^0$ will be

$$P_n = \langle n|\rho_{th}|n\rangle = \frac{e^{-\beta E_n^0}}{Z_0}. \tag{5}$$

Since the Hamiltonian, $\hat{H}_0$, given in Equation (2) is already diagonal in the usual Zeeman basis which diagonalizes $\hat{\mathbf{I}}_z$, Equation (4) can be rewritten in matrix form as

$$\rho_{th} = \frac{1}{Z_0} \begin{pmatrix} e^{\frac{\omega_0}{T}} & 0 & 0 \\ 0 & 1 & 0 \\ 0 & 0 & e^{\frac{-\omega_0}{T}} \end{pmatrix}.$$

The partition function is the trace of this matrix,

$$Z_0 = tr(e^{\frac{-\hat{H}_0}{T}}) = 1 + e^{\frac{\omega_0}{T}} + e^{\frac{-\omega_0}{T}}. \tag{6}$$

Using the hyperbolic trigonometric relations of the form: $e^{\pm x} = \frac{1 \pm \tanh(\frac{x}{2})}{1 \mp \tanh(\frac{x}{2})}$, the thermal density matrix can be rewritten in a convenient way as

$$\rho_{th} = \frac{1}{3 + f^2} \begin{pmatrix} 1 + 2f + f^2 & 0 & 0 \\ 0 & 1 - f^2 & 0 \\ 0 & 0 & 1 - 2f + f^2 \end{pmatrix} \tag{7}$$

where we have used $f = \tanh(\frac{x}{2})$ and $x = \frac{\omega_0}{T}$

*3.2. Spin in a Rotating Magnetic Field*

Immediately after the first energy measurement, we initiate the r.f. field

$$\mathbf{B}_1(t) = B_1(\cos \omega_z t \hat{x} + \sin \omega_z t \hat{y}). \tag{8}$$

where $\omega_z$ may be positive or negative. Although $B_1 \ll B_0$, it plays the role to tip the magnetization away from the z-axis into the xy-plane giving rise to the nuclear magnetic resonance (NMR) signal in the form of an induced voltage in an orthogonal plane.

Our system under the collective action of the strong static field, $\mathbf{B}_0$ and an alternating weak radio frequency field, $\mathbf{B}_1(t)$ can be described as

$$\mathbf{B}(t) = B_0 \hat{z} + B_1(\cos \omega_z t \hat{x} + \sin \omega_z t \hat{y}). \tag{9}$$

The magnetic spin Hamiltonian (which describes the way the nuclear magnetic energy changes as the nuclei rotate) will be

$$\begin{aligned} \hat{H}(t) &= -\boldsymbol{\mu} \cdot \mathbf{B} \\ &= -\omega_0 \hat{\mathbf{I}}_z - \omega_1(\hat{\mathbf{I}}_x \cos \omega_z t + \hat{\mathbf{I}}_y \sin \omega_z t) \end{aligned} \tag{10}$$
$$\hat{H}(t) = \hat{H}_0 + \hat{H}_1(t) \tag{11}$$

where we have used $\omega_0 = \gamma B_0$ and $\omega_1 = \gamma B_1$. Thus, the total Hamiltonian of the sample has a time-independent z-component, $H_0 = -\omega_0$, and a circularly polarized field representing a magnetic field rotating in the xy-plane, $H_1(t) = -\omega_1(\cos \omega_z t + \sin \omega_z t)$.

Now, by using the matrix form of nuclear spin components for the spin-1 system, $\hat{I}_z$, $\hat{I}_x$, and $\hat{I}_y$ the matrix representation of the above time-dependent Hamiltonian in Equation (10) will become

$$\hat{H}(t) = \begin{pmatrix} -\omega_0 & -\frac{\omega_1}{\sqrt{2}}(\cos \omega_z t - i \sin \omega_z t) & 0 \\ -\frac{\omega_1}{\sqrt{2}}(\cos \omega_z t + \sin \omega_z t) & 0 & -\frac{\omega_1}{\sqrt{2}}(\cos \omega_z t - i \sin \omega_z t) \\ 0 & -\frac{\omega_1}{\sqrt{2}}(\cos \omega_z t + i \sin \omega_z t) & \omega_0 \end{pmatrix}. \tag{12}$$

We note here that as a result of this time-dependent field, the system evolves in time.

To study the non-equilibrium properties of the system, we must know the initial thermal state, $\rho_{th}$, given in Equation (2) and the time-evolution operator, $\hat{U}(t)$ given as

$$\psi(t) = \hat{U}(t)|n\rangle; \hat{U}(t=0) = \mathbb{1}. \tag{13}$$

where $|n\rangle$ represents one of the three possible states of the spin-1 system $|1,1\rangle$ or $|1,0\rangle$ or $|1,-1\rangle$.

The Schrodinger equation corresponding to this Hamiltonian, $\hat{H}(t)$, can be given as

$$\begin{aligned} i\partial_t \hat{U}(t) &= \hat{H}(t)\hat{U}(t) \\ i\partial_t \hat{U}(t) &= \{-\omega_0 \hat{I}_z - \omega_1(\hat{I}_x \cos \omega_z t + \hat{I}_y \sin \omega_z t)\}\hat{U}(t) \end{aligned} \tag{14}$$

where we have used $\partial_t = \frac{\partial}{\partial_t}$. Substituting the value of $\hat{H}(t)$ from Equation (12) into Equation (14) yields

$$i\frac{\partial \hat{U}(t)}{\partial t} = \begin{pmatrix} -\omega_0 & -\frac{\omega_1}{\sqrt{2}}e^{-i\omega_z t} & 0 \\ -\frac{\omega_1}{\sqrt{2}}e^{i\omega_z t} & 0 & -\frac{\omega_1}{\sqrt{2}}e^{-i\omega_z t} \\ 0 & -\frac{\omega_1}{\sqrt{2}}e^{i\omega_z t} & \omega_0 \end{pmatrix} U(t). \tag{15}$$

where we have used the Euler formula: $e^{\pm i\omega_z t} = \cos \omega_z t \pm i \sin \omega_z t$.

For the spin-1 system, the unitary time evolution operator, $\hat{U}(t)$, can be expressed in matrix form (by using Zeeman basis) as

$$\hat{U}(t)|n\rangle = \begin{pmatrix} \hat{U}_+(t) \\ \hat{U}_0(t) \\ \hat{U}_-(t) \end{pmatrix}. \tag{16}$$

Then, by substituting Equation (16) into Equation (15) and performing matrix multiplication it yields:

$$i\begin{pmatrix} \partial_t \hat{U}_+(t) \\ \partial_t \hat{U}_0(t) \\ \partial_t \hat{U}(t) \end{pmatrix} = \begin{pmatrix} -\omega_0 \hat{U}_+(t) - \frac{\omega_1}{\sqrt{2}}\hat{U}_0(t)e^{-i\omega_z t} \\ -\frac{\omega_1}{\sqrt{2}}\hat{U}_+(t)e^{i\omega_z t} - \frac{\omega_1}{\sqrt{2}}\hat{U}_-(t)e^{-i\omega_z t} \\ -\frac{\omega_1}{\sqrt{2}}\hat{U}_0(t)e^{i\omega_z t} + \omega_0 \hat{U}_-(t). \end{pmatrix}. \tag{17}$$

This leads to the three coupled equations

$$\begin{aligned} (1) \quad i\partial_t \hat{U}_+(t) &= -\omega_0 \hat{U}_+(t) - \frac{\omega_1}{\sqrt{2}}\hat{U}_0(t)e^{-i\omega_z t} \\ (2) \quad i\partial_t \hat{U}_0(t) &= -\frac{\omega_1}{\sqrt{2}}(\hat{U}_+(t)e^{i\omega_z t} + \hat{U}_-(t)e^{-i\omega_z t}) \\ (3) \quad i\partial_t \hat{U}_-(t) &= -\frac{\omega_1}{\sqrt{2}}\hat{U}_0(t)e^{i\omega_z t} + \omega_0 \hat{U}_-(t). \end{aligned} \tag{18}$$

The time-dependence in the coefficients of these three equations can be eliminated by defining a new time-varying operator by the transformation method. This transformation corresponds to going to a coordinate system that rotates with an angular frequency $\omega_z$ about the $z$-axis

$$\hat{U}'(t) = e^{i\omega_z t \hat{I}_z}\hat{U}(t) \implies \hat{U}(t) = e^{-i\omega_z t \hat{I}_z}\hat{U}'(t). \tag{19}$$

Accordingly, the three coupled equations in Equation (18) can be rewritten as

$$(1) \quad i\partial_t \hat{U}'_+(t) \;=\; -(\omega_0 + \omega_z)\hat{U}'_+(t) - \frac{\omega_1}{\sqrt{2}}\hat{U}'_0(t) \tag{20}$$

$$(2) \quad i\partial_t \hat{U}'_0(t) \;=\; -\frac{\omega_1}{\sqrt{2}}\{\hat{U}'_+(t) + \hat{U}'_-(t)\} \tag{21}$$

$$(3) \quad i\partial_t \hat{U}'_-(t) \;=\; (\omega_0 + \omega_z)\hat{U}'_-(t) - \frac{\omega_1}{\sqrt{2}}\hat{U}'_0(t). \tag{22}$$

As a result, the modified Schrodinger equation (in terms of $\hat{I}_z$ and $\hat{I}_x$) will become

$$i\partial_t \hat{U}'(t) = \{-(\omega_0 + \omega_z)\hat{I}_z - \omega_1 \hat{I}_x\}\hat{U}'(t). \tag{23}$$

Hence, the modified quantum mechanical time-independent Hamiltonian (in angular frequency units) for the spin-1 nuclei placed in an external magnetic field consisting of a static field, $B_0$ along the $z$-axis and a weak radio frequency (r.f) field, of amplitude $B_1$, polarized along the $x$-axis will be

$$\hat{H}' = \{-(\omega_0 - \omega)\hat{I}_z - \omega_1 \hat{I}_x\} \tag{24}$$

Note that we are considering the case where the oscillating r.f field is rotating along with the precessing spin (clockwise direction, $\omega_z = -\omega$) and will partake in resonance.

The solution to Equation (23) will be

$$\hat{U}'(t) = \hat{U}'(0)e^{-i\hat{H}'t} = e^{-i\hat{H}'t} \qquad \implies \hat{U}(t) = e^{-i\omega_z t \hat{I}_z}e^{-i\hat{H}'t} \tag{25}$$

where for unitary operator $\hat{U}(t=0) = \hat{U}'(t=0) = \mathbb{1}$.

Physically, Equation (24) states that in the rotating frame, the moment acts as though it effectively experienced a static magnetic field, $H_{\text{eff}}$. The moment therefore precesses in a cone of fixed angle $\theta$ about the direction of $H_{\text{eff}}$ at an angular frequency $\gamma H_{\text{eff}}$. The situation is illustrated in Figure 1 for a magnetic moment initially oriented along the $z$-direction.

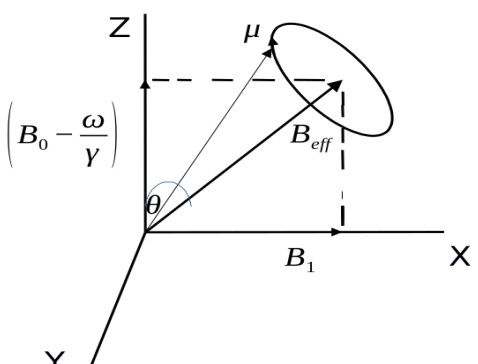

**Figure 1.** The precession of the nuclei spin under the effective field.

From Equation (23) we understood that the time-dependence of the externally applied radio frequency field, $\mathbf{B}_1(t)$, or its Hamiltonian, $\hat{H}_1(t)$, has been eliminated. In fact, we recognize it as representing the coupling of the nuclei spin with an effective field, $\mathbf{B}_{\text{eff}}$. So

the nuclei spin acts as though it experiences effectively a magnetic field $B_{\text{eff}}$ which can be defined mathematically as

$$B_{\text{eff}} = \Omega = \sqrt{B_1^2 + (B_0 - \frac{\omega}{\gamma})^2}. \tag{26}$$

The angle $\theta$ can be obtained from the components of the effective field as

$$\theta = \tan^{-1}\left(\frac{B_1}{(B_0 - \frac{\omega}{\gamma})}\right). \tag{27}$$

As a result, the corresponding effective energy operator, $\hat{H}'$, can be expressed in terms of $\theta$ as

$$\hat{H}' \;=\; -\gamma\Omega(\hat{I}_z \cos\theta + \hat{I}_x \sin\theta). \tag{28}$$

This is the polar form representation of the modified Hamiltonian, $\hat{H}'$; where $\theta$ is an angle between the effective field and the axis of rotation (z-axis).

Now, by substituting this form of $\hat{H}'$ in Equation (28) into Equation (25) one can obtain the full time evolution operator, $\hat{U}(t)$, in matrix form as

$$\hat{U}(t) \;=\; \begin{pmatrix} e^{i\omega_z t} & 0 & 0 \\ 0 & 1 & 0 \\ 0 & 0 & e^{-i\omega_z t} \end{pmatrix} e^{i\alpha\hat{M}} \tag{29}$$

where we have used $\alpha = \gamma\Omega t$ and $\hat{M} = \hat{I}_z \cos\theta + \hat{I}_x \sin\theta$. The operator $\hat{M}$ satisfies the conditions: $\hat{M} = \hat{M}^{2n+1}$ and $\hat{M}^2 = \hat{M}^{2n}$. Whenever this is true, a direct Taylor series expansion of $e^{-i\hat{H}'} = e^{i\alpha\hat{M}}$ where $\alpha$ is a constant, gives

$$e^{i\alpha\hat{M}} = \hat{1} + i\hat{M}\sin\alpha - (1 - \cos\alpha)\hat{M}^2. \tag{30}$$

Then, after solving for the matrix form of $\hat{M}$, and $\hat{M}^2$ and substituting their value in Equation (30) one can obtain the full time evolution operator, $\hat{U}(t)$, as

$$\hat{U}(t) = \begin{pmatrix} \nu(t) & -e^{-i\omega_z t}\nu^*(t) & -\chi^*(t) \\ -\nu^*(t) & 1 - 2e^{-i\omega_z t}\chi(t) & \nu(t) \\ -\chi(t) & e^{i\omega_z t}\nu(t) & \nu^*(t) \end{pmatrix}. \tag{31}$$

where we have defined

$$\nu(t) = e^{-i\omega_z t}\left\{\frac{\sin^2\theta}{2} + \left(1 - \frac{\sin^2\theta}{2}\right)\cos\alpha + i\cos\theta\sin\alpha\right\}; \tag{32a}$$

$$v(t) = \frac{\sin\theta}{\sqrt{2}}\left(\cos\theta(1 - \cos\alpha) + i\sin\alpha\right); \tag{32b}$$

$$\chi(t) = e^{i\omega_z t}\left(\frac{\sin^2\theta}{2}(1 - \cos\alpha)\right). \tag{32c}$$

To obtain a better physical interpretation of Equation (31), let us consider the situation where the system initially starts in the pure state (eigenstate) $|1,1\rangle$ of $\hat{I}_z$. Then, the probability that after a time 't' the system will be found in state $|1,0\rangle$ will be

$$\begin{aligned} Prob._{|+1\rangle \to |0\rangle} &= \left|\langle 1,0|U(t)|1,1\rangle\right|^2 \\ &= v(t)v^*(t) = \left|v(t)\right|^2. \end{aligned} \tag{33}$$

Then, by substituting for $v(t)$ from Equation (32) we obtain

$$Prob._{|+1\rangle \to |0\rangle} = \sin^2 \theta (1 - cos\alpha) - \frac{\sin^4 \theta}{2}(1 - cos\alpha)^2. \tag{34}$$

Similarly, the probability that the system of spin-1 nuclei that was in state $|1, 1\rangle$ will be found in state $\langle 1, -1|$ can be given as

$$Prob._{|+1\rangle \to |-1\rangle} = \left| \chi(t) \right|^2 = \frac{\sin^4 \theta}{4}(1 - \cos \alpha)^2. \tag{35}$$

Moreover, the unitary condition

$$U^\dagger(t)U(t) = \mathbb{1}. \tag{36}$$

implies that $|v(t)|^2 + |v(t)|^2 + |\chi(t)|^2 = 1$. This further gives $|v(t)|^2 = 1 - \{|v(t)|^2 + |\chi(t)|^2\}$, which represents the probability that no transition occurs. From Equation (32) we have seen that $v(t), v(t), \chi(t)$ are all the functions of $\sin \theta$. Therefore, this attributes a physical meaning to the angle $\theta$, defined in Equation (27) as representing the transition probability.

In fact, at resonance, where $\omega_z = -\gamma B_0$ or $\omega = \gamma B_0$ we have: $\sin \theta = 1$ and $\cos \theta = 0$. Hence, at resonance Equation (32) can be rewritten as:

$$v(t) = \frac{e^{-i\omega_z t}}{2}[1 + \cos \alpha] \implies v^*(t) = \frac{e^{i\omega_z t}}{2}[1 + \cos \alpha] \tag{37a}$$

$$v(t) = \frac{i}{\sqrt{2}} \sin \alpha \implies v^*(t) = \frac{-i}{\sqrt{2}} \sin \alpha \tag{37b}$$

$$\chi(t) = \frac{e^{i\omega_z t}}{2}(1 - \cos \alpha) \implies \chi^*(t) = \frac{e^{-i\omega_z t}}{2}(1 - \cos \alpha) \tag{37c}$$

For further discussion of the dynamics taking place as the nuclei of spin-1 interact with external magnetic field, having obtained the initial density matrix, $\rho_{th}$, and the full time-evolution operator, $\hat{U}(t)$, it is necessary to study spin polarization. The evolution of any observable A is given as

$$\langle A \rangle_{th} = tr\{\hat{U}^\dagger(t)A\hat{U}(t)\rho_{th}\}. \tag{38}$$

## 4. Dynamics of the Spin-One System

In this study, the strong static magnetic field, $B_0\hat{z}$, is used to split the energy of the nuclei into three energy levels, the energy-level diagram for a spin-1 nucleus **I** (as shown in Figure 2), therefore, has three energy levels, spaced evenly by $\omega_0 = \gamma B_0$ in natural units, if the quadrupole interaction is ignored.

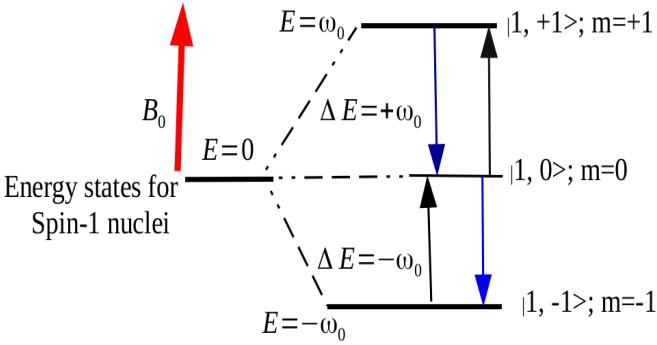

**Figure 2.** The representation of energy levels of spin-1 nuclei.

To illustrate the physics behind Equation (31), let us examine the time evolution of $\langle \hat{I}_x \rangle$, $\langle \hat{I}_y \rangle$, $\langle \hat{I}_z \rangle$. The evolution of the mean polarization of the nuclei of the spin-1 system can be calculated following Equation (38) as

$$\langle \hat{I}_i \rangle = tr\{U^\dagger(t)\hat{I}_i U(t)\rho_{th}\}. \tag{39}$$

where $\rho_{th}$ is the thermal density matrix (Equation 7) and $\hat{U}(t)$ is the time evolution operator (Equation (31)). For instance, we can compute the evolution of the mean polarization components $\langle \hat{I}_i \rangle$, where $i = x, y, z$. All calculations are reduced to the multiplication of $3 \times 3$ matrices. Accordingly, the mean polarization component in the $z$-direction will be

$$
\begin{aligned}
\langle \hat{I}_z \rangle &= tr\{U^\dagger(t)\hat{I}_z U(t)\rho_{th}\} \\
&= \frac{4f}{3+f^2}\left(|\nu|^2 - |\chi|^2\right) = \frac{4f}{3+f^2}\left\{1 - \sin^2\theta(1 - \cos\Omega t\gamma)\right\}.
\end{aligned} \tag{40}
$$

In the same manner, the mean polarization along the x and y-axis will be

$$\langle \hat{I}_x \rangle = \frac{4f\sin\theta}{(3+f^2)}\left\{\cos\theta\cos\omega_z t(1 - \cos\alpha) - \sin\alpha\sin\omega_z t\right\}. \tag{41}$$

$$\langle \hat{I}_y \rangle = \frac{4f\sin\theta}{(3+f^2)}\left\{\cos\theta\cos\omega_z t(1 - \cos\alpha) + \sin\alpha\sin\omega_z t\right\}. \tag{42}$$

At resonance, where $\sin\theta = 1$, they become

$$\langle \hat{I}_z \rangle = \frac{4f}{3+f^2}\cos(\Omega t\gamma). \tag{43}$$

$$\langle \hat{I}_x \rangle = \frac{-4f}{(3+f^2)}\left\{\sin\alpha\sin\omega_z t\right\}. \tag{44}$$

$$\langle \hat{I}_y \rangle = \frac{4f}{(3+f^2)}\left\{\sin\alpha\sin\omega_z t\right\}. \tag{45}$$

These equations (Equations (43)–(45)) describe a parametric curve in a sphere of radius, $R = \frac{4f}{3+f^2}$, which is the initial magnetization.

## 5. Properties of Work Distribution

In the first Section 5.1, we evaluate the nine possible work distributions and their corresponding probabilities. In Section 5.2, using $P(W)$, the average work is evaluated and its behavior is studied under different conditions. Lastly, the average work as a function of time is explored in Section 5.3.

### 5.1. The Work Distribution

The work performed in the non-equilibrium transformation process is the difference between the energy measurements made at final and initial states

$$W = E_f - E_i. \tag{46}$$

For our specific case of spin-1 nuclei, which is the three-state system, we figure out the possible work values in energy transformation. At the first instance of time (t = 0), the Hamiltonian was $H_i = -\omega_0\hat{I}_z$ which further tells us that the initial energy eigenvalues were $E^i_{-1} = \omega_0$ in the state $|-1\rangle$, $E^i_0 = 0$ in the state $|0\rangle$ and $E^i_{+1} = -\omega_0$ in the state $|+1\rangle$. Then, after some arbitrary time 't' the Hamiltonian became $H_f = -\omega_0\hat{I}_z + B_1(\cos\theta\hat{I}_x + \sin\theta\hat{I}_y)$. This gives the final instantaneous energy eigenvalues of $E^f_\pm = \mp\sqrt{B_0^2 + B_1^2(t)}$ in states $|\pm 1\rangle$, and $E_0 = 0$ in state $|0\rangle$. However, since $B_1 \ll B_0$,

the final eigenvalues are very similar to the initial eigenvalues. Consequently, the three values of $W$ are very close to zero. To simplify the discussion, let us suppose that the radio frequency field $B_1(t) = B_1 (\cos \omega_z t, \sin \omega_z t, 0)$ always changes by a full period. That is, we assume that the final protocol time $\tau$ is given by

$$\tau = \frac{2\pi l}{\omega}, l = 1, 2, 3, \cdots. \tag{47}$$

Physically, this tells us that for $\omega$ being a very fast frequency, we measure the work $W$ after a certain amount of complete cycles. In this case, $H_f = H_i$, which further implies that both the two measurements (the initial and final) may have the same energy spectrum: $E\pm = \mp\omega_0$ and $E_0 = 0$ in their respective states. Thus, we can obtain seven possibilities of work distributions for spin-1 nuclei in an effective magnetic field. These are:

$$
\begin{aligned}
&(1) \quad W = E_{+1} - E_{-1} = -2\omega_0 \Longrightarrow |-1\rangle \rightsquigarrow |+1\rangle \\
&(2) \quad W = E_{-1} - E_{+1} = 2\omega_0 \Longrightarrow |+1\rangle \rightsquigarrow |-1\rangle \\
&(3) \quad W = E_0 - E_{+1} = \omega_0 \Longrightarrow |+1\rangle \rightsquigarrow |0\rangle \\
&(4) \quad W = E_{+1} - E_0 = -\omega_0 \Longrightarrow |0\rangle \rightsquigarrow |+1\rangle \\
&(5) \quad W = E_{-1} - E_0 = \omega_0 \Longrightarrow |0\rangle \rightsquigarrow |-1\rangle \\
&(6) \quad W = E_0 - E_{-1} = -\omega_0 \Longrightarrow |-1\rangle \rightsquigarrow |0\rangle \\
&(7) \quad W = 0 \Longrightarrow |0\rangle \rightsquigarrow |0\rangle \text{or} |+1\rangle \rightsquigarrow |+1\rangle \text{or} |-1\rangle \rightsquigarrow |-1\rangle.
\end{aligned} \tag{48}
$$

As we can understand from this illustration, the work required to flip the nuclei spin from its initial state of $|-1\rangle$ to $|+1\rangle$ is equal to $-2\omega_0$ while the work required to flip in the reverse is $2\omega_0$. On the other hand, the work performed to flip the nuclei spin from initial states of $|0\rangle$ to the final state of $|+\rangle$ is equal to $-\omega_0$ and that performed in the reverse is $\omega_0$. Likewise, the work performed to flip the spin from initial states of $|-1\rangle$ to the final state of $|0\rangle$ is equal to $-\omega_0$ while that performed in the reverse is $\omega_0$. The last three distributions $(7, 8, 9)$ correspond to the case where there is no flip at all. So, here in our consideration of the spin-1 system there are seven possibilities of the work distribution unlike that of reference [19].

Now, the probability of these work distributions can be readily computed by using

$$P(W) = \sum p_n^0 p_{m|n}^{\tau} \delta \left( W - (E_m^\tau - E_n^0) \right). \tag{49}$$

Accordingly, using Equations (7) and (32) the probabilities for the work distributions in Equation (48) are obtained as

$$
\begin{aligned}
&(1) \quad P(W = -2\omega_0) = \frac{1 - 2f + f^2}{3 + f^2} |\chi(t)|^2 \Longrightarrow |-1\rangle \rightsquigarrow |+1\rangle \\[2mm]
&(2) \quad P(W = 2\omega_0) = \frac{1 + 2f + f^2}{3 + f^2} |\chi(t)|^2 \Longrightarrow |+1\rangle \rightsquigarrow |-1\rangle \\[2mm]
&(3) \quad P(W = \omega_0) = \frac{1 + f}{3 + f^2} |v(t)|^2 \Longrightarrow |+1\rangle \rightsquigarrow |0\rangle \quad \text{or} \quad |0\rangle \rightsquigarrow |-1\rangle \\[2mm]
&(4) \quad P(W = -\omega_0) = \frac{1 - f}{3 + f^2} |v(t)|^2 \Longrightarrow |0\rangle \rightsquigarrow |+1\rangle \quad \text{or} \quad |-1\rangle \rightsquigarrow |0\rangle \\[2mm]
&(5) \quad P(W = 0) = \frac{(1 - f^2)}{3 + f^2} s(t) \Longrightarrow |0\rangle \rightsquigarrow |0\rangle \\[2mm]
&(6) \quad P(W = 0) = \frac{(1 + f^2)}{3 + f^2} |v(t)|^2 \Longrightarrow |+1\rangle \rightsquigarrow |+1\rangle \quad \text{or} \quad |-1\rangle \rightsquigarrow |-1\rangle.
\end{aligned} \tag{50}
$$

where $s(t) = \left( 1 + 4|\chi(t)|^2 - 2e^{-i\omega_z t}\chi(t) - 2e^{i\omega_z t}\chi^*(t) \right)$.

If $\omega_0 > 0$ it is more likely that the spin will be aligned parallel to the field. In such a case, $f > 0$ and this further adds $P(W = 2\omega_0) > P(W = -2\omega_0)$ and $P(W = \omega_0) > P(W = -\omega_0)$. This implies that it is more likely that the field will promote a flip from $|+1\rangle$ to $|-1\rangle$ or from $|+1\rangle$ to $|0\rangle$ (and from $|0\rangle$ to $|-1\rangle$) than the other way around.

*5.2. The Average Work*

Since we already explicitly obtained $P(W)$, it is possible to compute the average work, $\langle W \rangle$, from the definition

$$\langle W \rangle = \sum_w w P(W = w). \tag{51}$$

Thus, for the work distribution given in Equation (48) and the corresponding probability distributions given in Equation (50), the average work will be

$$\langle W \rangle = \frac{2\omega_0 f}{3 + f^2} \left\{ v(t)|^2 + |\chi(t)|^2 \right\}. \tag{52}$$

In cases where $P(W)$ is not explicitly known, the average work performed during the transformation can be computed by using

$$\langle W \rangle = \langle \hat{H}_f \rangle_{t=\tau} - \langle \hat{H}_i \rangle_{t=0}. \tag{53}$$

In computing the expectation values of quantities related to the energy of the system, we may always use the unperturbed case (where a nucleus of spin-1 is only in the static, strong and uniform magnetic field, $\hat{B}_0$), for the reason that we expect the result, which we match with what we know from measurements in the unperturbed condition. Particularly, in our case of the spin-1 system in a strong static magnetic field, $\mathbf{B}_0$ (applied in the direction of the z-axis) and where a weak alternating magnetic field, $\mathbf{B}_{1,\prime}$ is applied along the perpendicular direction to the axis of rotation (z-axis) we use $\hat{H}_0 = -\omega_0 \hat{I}_z$ as a Hamiltonian operator in unperturbed conditions.

Accordingly, the mean value of the Hamiltonian in this case will be

$$\langle \hat{H}_0 \rangle = \langle -\omega_0 \hat{I}_z \rangle = -\omega_0 \langle \hat{I}_z \rangle. \tag{54}$$

Then, by solving for $\langle \hat{I}_z \rangle$ at initial measurement ($t = 0$) and at final measurement ($t = \tau$), the average energy at any instant of time can be simply given as

$$\begin{aligned} \langle W \rangle &= \langle \hat{H}_0 \rangle_{t=\tau} - \langle \hat{H}_0 \rangle_{t=0} \\ &= \frac{4f\omega_0}{3 + f^2} \sin^2 \theta \left\{ 1 - \cos(\Omega\tau) \right\}. \end{aligned} \tag{55}$$

Now, by substituting for $\sin \theta = \frac{B_1}{\Omega}$, the average work at a time 't' will be given as

$$\langle W \rangle = \frac{4f\omega_0}{3 + f^2} \frac{B_1^2}{\Omega^2} \left\{ 1 - \cos(\Omega\tau) \right\}. \tag{56}$$

The average work, therefore, oscillates indefinitely with frequency, $\Omega$. This is the consequence of the fact that the evolution operator is unitary.

The amplitude multiplying the average work is proportional to the initial magnetization, $R = \frac{f}{3+f^2}$, and to the ratio, $\frac{B_1^2}{\Omega^2}$. This looks like the known Lorentzian function, and it represents a sharp peak at the resonance frequency ($\omega = \omega_0$), which becomes sharper for smaller value of $B_1$. The maximum possible work, therefore, occurs at resonance and has the value

$$\langle W \rangle = \frac{4\omega_0 f}{3 + f^2}. \tag{57}$$

Now, by using the assumption in Equation (47), the average work in Equation (56) can be expressed as:

$$\langle W \rangle = \frac{4 f \omega_0}{3 + f^2} \frac{B_1^2}{\Omega^2} \left\{ 1 - \cos(\frac{2\pi l}{\omega}\Omega) \right\}. \tag{58}$$

This result is something which is similar to the result obtained in the case of spin-half studied in reference [19], despite the additional multiplicative term of $\left\{ 1 - \cos(\frac{2\pi l}{\omega}\Omega) \right\}$. By substituting for $\Omega = \sqrt{(\omega - B_0)^2 + B_1^2}$ and rearranging terms, the full expression of the mean work in Equation (58) will be given as:

$$\frac{\langle W \rangle}{4 R B_0} = \left( \frac{K_1}{K_3} \right)^2 \left\{ 1 - \cos(\frac{2\pi l K_3}{K_2}) \right\}. \tag{59}$$

where we have used the notations:

$$K_1 = \frac{B_1}{B_0}; K_2 = \frac{\omega}{B_0}; k_3 = \sqrt{(K_2 - 1)^2 + K_1^2}; R = \frac{f}{3 + f^2}. \tag{60}$$

This result in Equation (60) is further illustrated in Figure 3 for different values of $l$ and for the fixed ratio of the externally applied fields, $\frac{B_1}{B_0}$. The resonance condition is achieved when $K_2 = \frac{\omega}{B_0} = 1$. As it is possible to understand from the figure, the dependence of $\langle W \rangle_l$ on the angular frequency $\omega$ is quite complicated, but it depends in a meaningful manner on the duration $l$ of the protocol implemented. In general, the work performed (conducted) increases sharply as it approaches the resonance condition.

For small values of $l$ (see Figure 3a–e), the average work tends to increase (as a function of angular frequency, $\omega$) to reach the resonance value, but soon it begins to smoothly decrease to reattain the initial equilibrium condition before it reaches the resonance value. On the other hand, for a large value of $l$ the average work sharply increases near resonance value, $\frac{\omega}{B_0} = 1$, and soon attains its maximum at resonance. This is explicitly shown in Figure 3f for $l = 1000$. In such a case, the oscillation becomes very fast as $\omega$ is varied.

Taking the energy relaxation time of the system to be of the order of 3 s one can estimate the upper bound in the number of repeated cycles for the radio frequency (rf) oscillations to perform without violating the approximation of the 'isolated system'. Assuming the rf oscillations to have a period as large as 1 ms, repeating the number of cyclic oscillations a thousand times will take a span of 1 s. This is definitely less than 3 s of time making our operation on the safe side [24,25].

Now, we can deduce the free energy of the system of spin-1 nuclei as follows. For our choice of time $\tau$ in Equation (47), since $H_f = H_i$ the change in free energy becomes $\Delta F = 0$ and this further implies that the free energy is a function only of the Hamiltonian, measured either at the initial state or final state. In such a case, Equation (59) is in agreement with $W \geq \Delta F$, which now reads $\langle W \rangle_l \geq 0$. Therefore, the average work is always greater than the free energy as expected. However, we must be careful of the individual realizations of the work distribution. For example, the cases such as $W = -\omega$ and $W = -2\omega$ do not fulfill the condition above ($W \geq \Delta F$). This further indicates that individual realizations may very well violate the second law, but the average work does not.

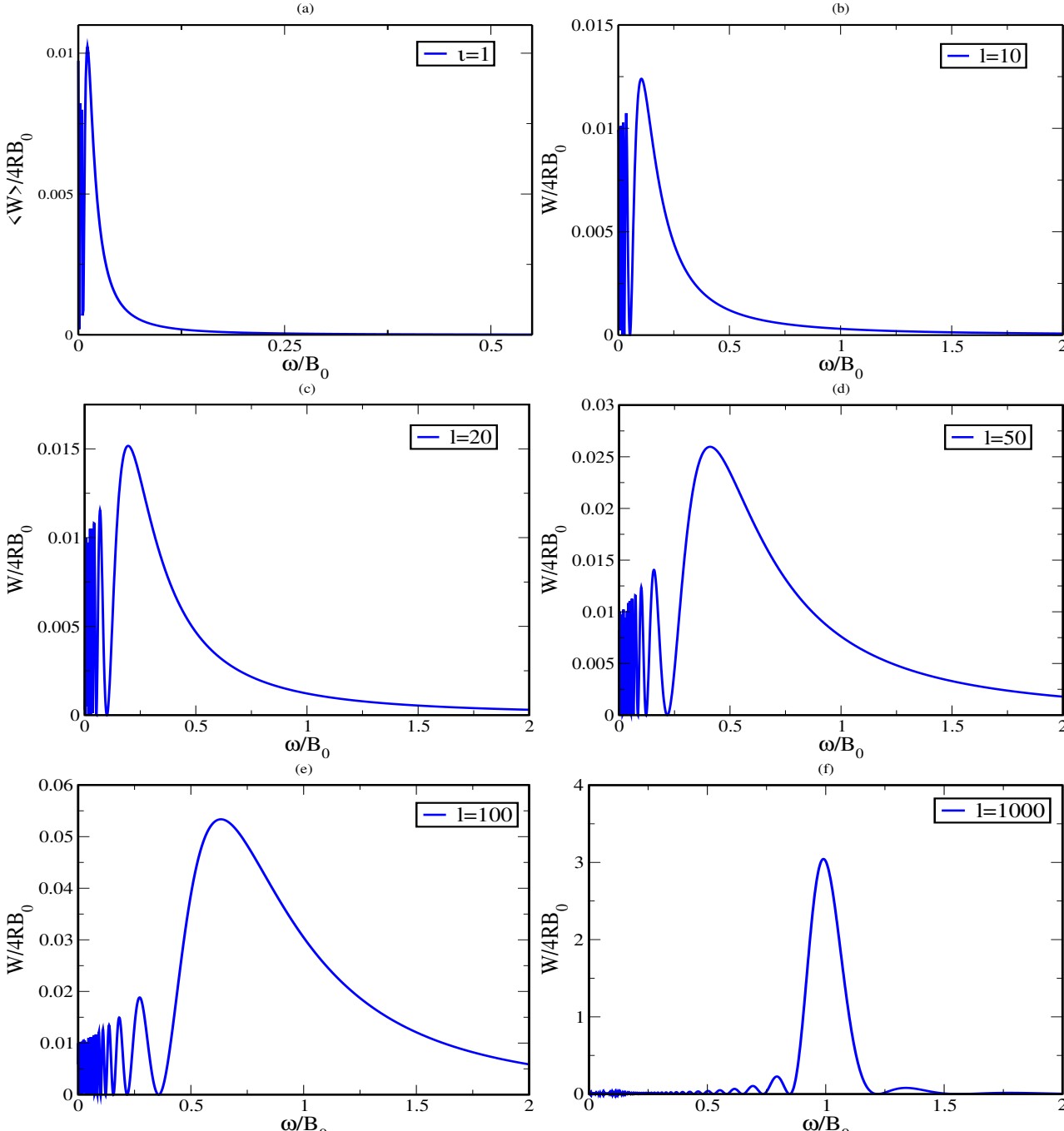

**Figure 3.** Average work $\frac{\langle W \rangle}{4RB_0}$ vs. $\frac{\omega}{B_0}$ computed using Equation (59) for fixed value $\frac{B_1}{B_z} = 0.01$. The average work, $\langle W \rangle_l$, depends sensibily on the duration 'l' of the protocol. For small values of 'l' (see (**a**)–(**c**)) it may be very small at resonance. But, for relatively large values of 'l' it increases close to resonance (see (**d**,**e**)). When 'l' is very large (l = 1000) a maximum is obtained exactly at resonance (see (**f**)). Note that the scales are different for each image.

### 5.3. The Average Work as a Function of Time

For completeness, we can also discuss the average work as a function of time, without making the assumption in Equation (47). The general formula for $\langle W \rangle_t$ is the difference between the average energy at time t (at which the protocol is switched off) and that of

the average energy at the initial time, t = 0 (just before the protocol was implemented). Therefore, by using Equations (10) and (54) the average work as a function of time will be

$$
\begin{aligned}
\langle W \rangle_t &= \langle H(t) \rangle_t - \langle H_0 \rangle_{t=0} \\
&= -B_0 \langle I_z \rangle_t - B_1 \{ \langle I_x \rangle \cos \omega t + \langle I_y \rangle \sin \omega t \} - (-B_0 \langle I_z \rangle_{t=0}).
\end{aligned} \tag{61}
$$

Now, by substituting for $\langle I_z \rangle$, $\langle I_x \rangle$, and $\langle I_y \rangle$ from Equations(40, 41 and 42), respectively, and simplifying the equation, we will have

$$
\begin{aligned}
\frac{\langle W \rangle}{4RB_0} = \left( \frac{K_1}{K_3} \right)^2 &\Bigg\{ \Big( 1 - \cos(B_0 K_3 t) \Big) - \frac{K_2 - 1}{K_3} \cos \omega t \Big( 1 - \cos(B_0 K_3 t) \Big) (\cos \omega t + \sin \omega t) \\
&+ \sin(B_0 K_3 t) \sin \omega t \Big( \sin \omega t - \cos \omega t \Big) \Bigg\}.
\end{aligned} \tag{62}
$$

This result is illustrated in Figure 4 for different values of our work parameter, $B_1(t)$ and the angular frequency, $\omega$, by which the spin is precessing around the resonance value, $\omega = \omega_0$. As we can see from the figure, the average work oscillates with two characteristic periods: a fast oscillation of frequency $\omega$ and a slow oscillation of frequency $\omega_1 = B_1$. The closer the oscillation is to $\omega_0$ the higher is the performance of the average work performed.

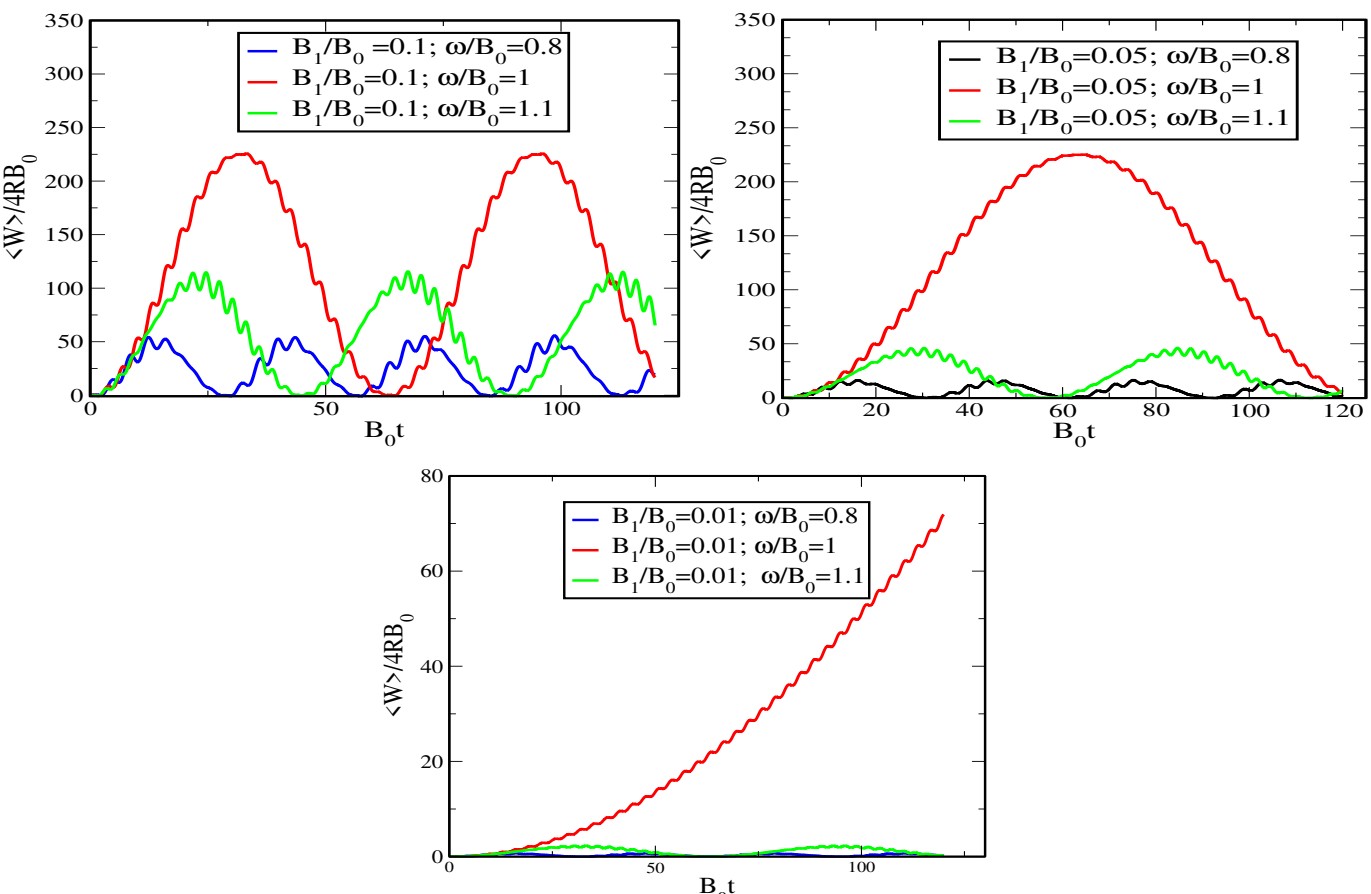

**Figure 4.** Average work computed by using Equation (62) $\left( \frac{\langle W \rangle}{4RB_0} \text{ vs. } B_0 t \right)$ for different values of $\frac{B_1}{B_0}$ and $\frac{\omega}{B_0}$.

## 6. Characteristic Function and Distribution of Work

The response of a quantum system (the nuclei of spin-1) to the perturbation by the external magnetic field can be further characterized by the change of energy contained in

the total system. Therefore, the characteristic function of the work distribution, which may comprise all aspects of the statistics of the work would be given as [17]

$$G(r) = tr\{\hat{U}^\dagger(\tau)e^{-ir\omega_0\hat{I}_z}\hat{U}(\tau)e^{ir\omega_0\hat{I}_z}\rho_{th}\}. \tag{63}$$

By substituting for $\hat{U}(t)$ from Equation (31) and for $\rho_{th}$ from Equation (7) into Equation (63) and going through long mathematical derivation of the characteristic function of work in a more simplified form is

$$
\begin{aligned}
G(r) \;=\; & \frac{1}{3+f^2}\Big\{2|v|^2(1+f^2) + s(t)(1-f^2) + 2|v|^2 e^{ir\omega_0}(1+f) \\
& +2|v|^2 e^{-ir\omega_0}(1-f) + |\chi|^2 e^{2ir\omega_0}(1+2f+f^2) + |\chi|^2 e^{-2ir\omega_0}(1-2f+f^2)\Big\}.
\end{aligned}
\tag{64}
$$

From the definition of the characteristic function we have that

$$G(r) = \langle e^{irW}\rangle = \int_{-\infty}^{\infty} P(W)e^{irW}dW. \tag{65}$$

Then, the power series expansion of this characteristic function, $G(r)$, can be expressed in terms of the statistical moments of work are as follows

$$G(r) = \langle e^{irW}\rangle = 1 + ir\langle W\rangle + \frac{(ir)^2}{2!}\langle W^2\rangle + \frac{(ir)^3}{3!}\langle W^3\rangle + \cdots. \tag{66}$$

where

$$\langle W^n\rangle = (-i)^n \frac{\partial^n G(r)}{\partial r^n}\Big|_{r=0}. \tag{67}$$

In particular, the first moment of the work would be

$$
\begin{aligned}
\langle W\rangle \;=\; & (-i)\frac{\partial G(r)}{\partial r}\Big|_{r=0} \\
=\; & \frac{\omega_0}{(3+f^2)}\Big\{2|v|^2(1+f) - 2|v|^2(1-f) + 2|\chi|^2(1+2f+f^2) - 2|\chi|^2(1-2f+f^2)\Big\}.
\end{aligned}
$$

Further simplification of this equation will yield

$$\langle W\rangle = \frac{4f\omega_0}{(3+f^2)}\Big\{|v|^2 + 2|\chi|^2\Big\}. \tag{68}$$

Then, by substituting for $v$ and $\chi$ from Equation (32) the first moment of the work will become

$$\langle W\rangle = \frac{4f\omega_0}{(3+f^2)}\sin^2\theta(1-\cos\alpha). \tag{69}$$

Again, substituting for $\alpha = \Omega t$, $\omega_0 = B_0$ and $\sin\theta = \frac{B_1}{\Omega}$ it can be rewritten as

$$\langle W\rangle = \frac{4fB_0}{(3+f^2)}\frac{B_1^2}{\Omega^2}\Big(1-\cos(\Omega t)\Big). \tag{70}$$

This is what we have already obtained in Equation (56)

The second moment of work, $\langle W^2\rangle$, can also be calculated as

$$
\begin{aligned}
\langle W^2\rangle \;=\; & \frac{\partial^2 G(r)}{\partial r^2}\Big|_{r=0} = (-i)\frac{\partial\langle W\rangle}{\partial r}\Big|_{r=0} \\
=\; & \frac{4\omega_0^2}{(3+f^2)}\Big(|v|^2 + 2|\chi|^2(1+f^2)\Big).
\end{aligned}
\tag{71}
$$

Again, substituting for $v$ and $\chi$ from Equation (32) we would have

$$\langle W^2 \rangle = \frac{4\omega_0{}^2}{(3+f^2)} \sin^2 \theta (1 - \cos \alpha) + \frac{4\omega_0{}^2 f^2}{(3+f^2)} \left( \frac{\sin^4 \theta}{2} (1 - \cos \alpha)^2 \right). \quad (72)$$

From the first moment of work obtained in Equation (69) one can find its mean square, $\langle W \rangle^2$, as

$$\langle W \rangle^2 = \frac{16 f^2 \omega_0^2}{(3+f^2)^2} \sin^4 \theta (1 - \cos \alpha)^2. \quad (73)$$

As a consequence, the variance of the work distribution becomes

$$\begin{aligned} Var(W) &= \langle W^2 \rangle - \langle W \rangle^2 \\ &= \frac{4\omega_0{}^2}{(3+f^2)} \left\{ \sin^2 \theta (1 - \cos \alpha) + f^2 \frac{(f^2 - 5)}{2(3+f^2)} \sin^4 \theta (1 - \cos \alpha)^2 \right\}. \quad (74) \end{aligned}$$

Finally, we compute the full distribution of work in terms of the work probability distribution, P(W), by using the characteristic function, $G(r)$, given in Equation (66). The probability distribution function is the inverse Fourier transform of the characteristic function, $G(r)$

$$P(W) = \frac{1}{2\pi} \int_{-\infty}^{\infty} dr\, G(r) e^{-irW}. \quad (75)$$

Thus, by substituting for $G(r)$ from Equation (64) into Equation (75) and by using the Dirac delta function of the form

$$\frac{1}{2\pi} \int_{-\infty}^{\infty} e^{ir(a-b)} dr = \delta(b - a) \quad (76)$$

the work probability distribution can be expressed as

$$\begin{aligned} P(W) = {} & \frac{1}{3+f^2} \left\{ 2|v|^2 (1+f^2) \delta(W) + \left( 1 + 4|\chi|^2 - 2\chi e^{-i\omega_z t} - 2\chi^* e^{i\omega_z t} \right) (1 - f^2) \delta(W) \right. \\ & + 2|v|^2 (1+f) \delta(W - \omega_0) + 2|v|^2 (1-f) \delta(W + \omega_0) \\ & \left. + |\chi|^2 (1 + 2f + f^2) \delta(W - 2\omega_0) + |\chi|^2 (1 - 2f + f^2) \delta(W + 2\omega_0) \right\}. \end{aligned} \quad (77)$$

Hence, the work interpreted as a random variable takes the following five distinct values: $W = 0, W = -\omega_0, W = \omega_0, W = -2\omega_0$, and $W = 2\omega_0$.

## 7. Summary and Conclusions

In this work, we took a very weakly interacting spin-1 magnetic system subjected to a strong static external magnetic field while in contact with a thermal bath. A predefined protocol was imposed on the system for a finite amount of time where its energy was recorded both at the start and finish times. Even though the interaction between the system and that of the measuring apparatus was unavoidable in a real system measurement scheme, here in this study, we used the method of the two-point measurement scheme [11]. Enough data enabled us to extract its statistical behavior such as work distributions and find the energy difference between the final and initial equilibrium energy values. In conclusion, we suggest that we take a collection of deuterium atomic nuclei subjected to undergo a cyclic process similar to what we have proposed, carry out the experiment and compare the result with our prediction.

**Author Contributions:** M.B. and M.M. developed the theoretical formalism and conceived of the presented idea. M.M. performed the analytic calculations and the numerical simulations. Both M.M. and Y.B. authors contributed to the final version of the manuscript. M.B. supervised the project. All

authors discussed the results and contributed to the final manuscript. All authors have read and agreed to the published version of the manuscript.

**Funding:** This research received no external funding.

**Data Availability Statement:** This manuscript has no associated data or the data will not be deposited.

**Acknowledgments:** We would like to thank The International Science Programme, Uppsala University, Uppsala, Sweden for the support they have provided to our research group. MM would like to thank Addis Ababa University and Wallaga University for financial support during his research work. YB would like to thank Addis Ababa University and Wolkite University for financial support during his research work.

**Conflicts of Interest:** We confess that this work is solely performed by three of us (M.M., M.B. and Y.B.) and we declare that there is no conflict of interest with any person regarding our manuscript.

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
