# Peer review of "Thermal and Quantum Fluctuation Effects on Non-Spherical Nuclei: The Case of Spin-1 System"

_condensedmatter, doi:10.3390/condmat7040062_

Round 1
Reviewer 1 Report
In this work, the authors analyze the energy distribution of the spin 1 system after the effect of a static (Zeeman) magnetic field along the z-axis and a transversal (fast) oscillating magnetic field.
They assume that the system at the initial time is either a pure eigenstate of the angular momentum along z either a quantum statistical distribution at thermal equilibrium. When the oscillating magnetic field is switched on, the authors assume unitary evolution, namely the spin is completely decoupled from the environment. This assumption allows them to calculate exactly the evolution of the system and to determine, in particular, the energy distribution of the spin conditioned with the initial state (pure state or thermal mixture).
Although I didn’t repeat the calculations, it seems to me that the mathematical calculations have carried out correctly and precisely.
I have three major comments:
1) Any real system is always coupled to the environment, i.e. the external bath. Therefore, the energy relaxation dynamics and the decoherence are unavoidable. If the system is extremely weakly coupled to the external worlds, the energy relaxation and the decoherence rate are proportionally very small such that, in the initial time range, one can neglect these processes – in a first approximation – and one can assume that the evolution is unitary. I suggest that the authors would comment this issue in the paper.
2) For the same reason, the time in the final results (or the number of cycles) cannot be assumed unbounded but, at some point, the approximation of “isolated system” is not valid anymore. So, increasing the number of cycles to 1000 – as shown in figure 3 – it requires that the total time is still small compared to the time scale of the energy relaxation and decoherence (i.e. the inverse of the associated rates).
Since the authors discuss the problem in a general (mathematical) way, without referring to some real experiments and putting real numbers, it is impossible to say if the assumptions made in the paper are valid or not. It would be very interesting whether the author could refer to some experiments or they can comment this issue.
3) The main theoretical result is the energy distribution of the spin conditioned with the initial state – a quantity that the authors refer to as “work”. This nomenclature is based on the assumption that an ideal (abstract) projective measurement of the quantum state of the system is performed leaving the state of the system in one of the eigenstates of the unperturbed Hamiltonian. Again, a real measurement scheme would imply to take the interaction of the system with the measure apparatus into account. Then aging, in the latter case, the time evolution not unitary anymore. Maybe the authors can shortly comment this issue in the paper.
In general, I found a bit missing the motivations of these calculations: what was the original question/issue/problem that the authors wanted to address? What is the usefulness of having these quantities computed? Why is what important to solve exactly the dynamics of the system? If one uses the simple Fermi’s Golden rule, one would obtain a different result?
Minor comments:
- there are some typos in the formulas and in the figures, for example
* the Eq.23 and Eq. 24, they differ by a sign in front of the frequency \omega
* in Eq.24, the subscript “z” is missing in the frequency
* in many frequencies appearing in the formulas, in the text and in the figure, the subscript is missing
* in Eq.76 a factor 2\pi is missing in the definition of the delta function
- can you include a citation before the formula Eq. 63? I guess that other persons in the past literature have derived this expression for the characteristic function.
Author Response
Response to Reviewer 1 Comments
[Major Comment 1] Any real system is always coupled to the environment, i.e. the external bath. Therefore, the energy relaxation dynamics and the decoherence are unavoidable. If the system is extremely weakly coupled to the external worlds, the energy relaxation and the decoherence rate are proportionally very small such that, in the initial time range, one can neglect these processes – in a first approximation – and one can assume that the evolution is unitary. I suggest that the authors would comment this issue in the paper.
Response-1: Thank you very much for your clear and constructive comment. Just as you have raised, we considered the case where our system is extremely very weakly coupled to the environment such that the under going process is assumed to be unitary. We have incorporated your comments in the paper as it can be seen on page-3, section-2, paragraph-1, line-13 .
[Major Comment 2] For the same reason, the time in the final results (or the number of cycles) cannot be assumed unbounded but, at some point, the approximation of “isolated system” is not valid anymore. So, increasing the number of cycles to 1000 – as shown in figure 3 – it requires that the total time is still small compared to the time scale of the energy relaxation and decoherence (i.e. the inverse of the associated rates).
Since the authors discuss the problem in a general (mathematical) way, without referring to some
real experiments and putting real numbers, it is impossible to say if the assumptions made in the
paper are valid or not. It would be very interesting whether the author could refer to some
experiments or they can comment this issue.
Response-2: Thank you very much. Of course the number of cycles must be bounded in order to keep in touch with the assumption of the “isolation of the system under study”. Taking the energy relaxation time of the system to be of the order of 3 seconds one can estimate the upper bound in the number of repeated cycles for the radio frequency (rf) oscillations to perform with out violating the approximation of ‘isolated system’. Assuming the rf oscillations to have a period as large as 1ms, repeating the number of cyclic oscillations for thousand times will take a span of 1 second. This is definitely less than 3 second of time making our operation on the safe side. We commented this issue in the manuscript as you can see in the paper page-13 , subsection-5.2 , paragraph-3, line-1.
[Major Comment 3] The main theoretical result is the energy distribution of the spin conditioned with the initial state– a quantity that the authors refer to as “work”. This nomenclature is based on the assumption that an ideal (abstract) projective measurement of the quantum state of the system is performed leaving the state of the system in one of the eigenstates of the unperturbed Hamiltonian. Again, a real measurement scheme would imply to take the interaction of the system with the measure apparatus into account. Then aging, in the latter case, the time evolution not unitary anymore. Maybe the authors can shortly comment this issue in the paper.
Response-3: Thank you very much for pointing out this idea. Exactly, as you said in a real measurement scheme the interaction between the system and the measuring apparatus is unavoidable. But we have employed the two point measurement scheme by assuming our system isolated from the external environment as it is used in many quantum experiments. We have commented this issue in the manuscript as you can see on page-17, section-7, paragraph-1, line-4.
[General Comment] In general, I found a bit missing the motivations of these calculations: what was the original question/issue/problem that the authors wanted to address? What is the usefulness of having these quantities computed? Why is what important to solve exactly the dynamics of the system? If one uses the simple Fermi’s Golden rule, one would obtain a different result?
Response: Thank you very much. Spin-1 nuclei such as deuterium atomic isotopes when exposed to an external magnetic field will orient in three possible discrete states. We want to study the quantum thermodynamics of the spin-1 nuclei by employing the small radio frequency field. To understand the dynamics taking place through the process, we solved for the work distribution. To the best of our knowledge, we have not seen any previous work dealing with its quantum thermodynamic properties. Its rich properties motivated us to address it. We have commented this issue in section-1, page-2, paragraph-6, line-7.
Regarding the suggested Femi’s Golden rule we have not dealt with it here. But we may consider it in our future study.
[Minor Comments]
- there are some typos in the formulas and in the figures, for example
* the Eq.23 and Eq. 24, they differ by a sign in front of the frequency \omega
Response: Thank you very much for the comment. Although it looks like the two equations are the same, they represent different equations: Eq.23 refers to the Schrödinger equation and Eq.24 refers to the Hamiltonian.
* in Eq.24, the subscript “z” is missing in the frequency
Response: Thank you for this suggestion. It would have been interesting to explore this aspect. The subscript “z” is missing in the frequency ω because we further set on that ωz = -ω. See the statements just below Eq.24, on page-6 of the manuscript.
* in many frequencies appearing in the formulas, in the text and in the figure, the subscript is
missing
Response: For the same reason as in the above, mostly we preferred to use for frequencies appearing in the formulas, in the text and in the figure, without the subscript.
* in Eq.76 a factor 2\pi is missing in the definition of the delta function
Response: Thanks very much. We agree with this and have incorporated your suggestion in the
manuscript. We have rewritten Eq.76 as can be seen on page-17.
- can you include a citation before the formula Eq. 63? I guess that other persons in the past
literature have derived this expression for the characteristic function.
Response: Thanks for your kind reminders. Accordingly, we put the citation just before the formula as you can see on page-15, the first paragraph of section-6, line number-4.

Reviewer 2 Report
The authors extend the model of NMR for spin-half sample nuclei, Ref.[30], to NMR of spin-one nuclei.
Comments:
1. I think that the reader would benefit if the authors would discuss the difference between the results of Ref.[30] and their results.
2. Are there any similarities in the present results and results of Ref.[30], and why these similarities (if any) exist despite the difference in the systems under study?
After the authors would address the above comments, I can recommend this manuscript for publication.
Reviewer 3 Report
The manuscript deals with a quantum-thermodynamic description of spin-1 nuclei based on the concepts of quantum and statistical mechanics. To this end the system is driven out of equilibrium and the full distribution of the work generated by the process is analyzed in detail. Thus, the study contributes to our current understanding of quantum thermodynamics. The manuscript is well written, presents interesting results and can, therefore, be published as it stands in the journal Condensed Matter.
The authors may be interested in the following reference:
J. Gemmer, M. Michel, and G. Mahler, Quantum Thermodynamics - Emergence of Thermodynamic Behavior Within Composite Quantum Systems, Second Edition (Springer, Berlin, 2009)
